# Circumventing Doxorubicin Resistance Using Elastin-like Polypeptide Biopolymer-Mediated Drug Delivery

**DOI:** 10.3390/ijms23042301

**Published:** 2022-02-19

**Authors:** Sonja Dragojevic, Lindsay Turner, Drazen Raucher

**Affiliations:** 1Division of Radiation Oncology, Mayo Clinic and Foundation, 200 First Street, SW, Rochester, MN 55905, USA; dragojevic.sonja@mayo.edu; 2Department of Cell and Molecular Biology, University of Mississippi Medical Center, 2500 North State Street, Jackson, MS 39216, USA; lturner8@umc.edu

**Keywords:** drug resistance, doxorubicin, drug delivery, elastin-like polypeptide

## Abstract

Although doxorubicin (dox), an anthracycline antibiotic, is widely used and effective in treating cancer, its treatment efficiency is limited by low blood plasma solubility, poor pharmacokinetics, and adverse side effects, including irreversible cardiotoxicity. Moreover, cancer cells often develop drug resistance over time, which decreases the efficacy of anti-cancer drugs, including dox. In this study, we examine a macromolecular drug delivery system for its ability to specifically deliver doxorubicin to cancer cells with and without drug resistance. This drug delivery system consists of a multi-part macromolecule, which includes the following: elastin-like polypeptide (ELP), cell penetrating peptide (CPP), a cleavable linker (releasing at low pH), and a derivative of doxorubicin. ELP is thermally responsive and improves drug solubility, while the CPP mediates cellular uptake of macromolecules. We compared cytotoxicity of two doxorubicin derivatives, where one is cleavable (DOXO) and contains a pH-sensitive linker and releases dox in an acidic environment, and the other is non-cleavable (ncDox) doxorubicin. Cytotoxicity, apoptosis, cell cycle distribution and mechanism of action of these constructs were tested and compared between dox-responsive MCF-7 and dox-resistant NCI/ADR cell lines. Dox delivered by the ELP construct is comparably toxic to both sensitive and drug resistant cell lines, compared to unconjugated doxorubicin, and given the pharmacokinetic and targeting benefits conveyed by conjugation to ELP, these biopolymers have potential to overcome dox resistance in vivo.

## 1. Introduction

The current treatment for patients with solid malignant tumors consists of multimodal therapy, including surgery, followed by radiotherapy and chemotherapy. However, the systemic chemotherapy approach has limited utility due to off-target damage to healthy tissues, resulting in increased mortality. In addition to systemic toxicity, one of the common causes for the failure of the standard cancer chemotherapies is development of multidrug resistance (MDR) [1,2].

Multidrug resistance (MDR) is a manifestation of cancer cells that involves resistance to various chemotherapeutic drugs due to the intrinsic and acquired expression of the ABC transporter known as p-glycoprotein (P-gp), multidrug resistance protein 1 (MDR1) or ATP-binding cassette sub-family B member 1 (ABCB1) [3]. MDR can be induced by the initial administration of a single drug, and it expands resistance to cover a wide array of other unrelated chemo drugs [4,5], which are diverse both structurally and mechanistically, such as cisplatin, daunorubicin, docetaxel, doxorubicin, and epirubicin.

Many studies have attempted to target and attenuate MDR via introducing different agents such as bexarotene, biricodar (VX710), dexrazoxane, ethacrynic acid, verapamil, valspodar (PSC833), and tariquidar (XR9576) [6]. These agents either inhibit or saturate the P-gp pumps to reduce drug resistance. In addition to preclinical efforts to address MDR, some of the agents mentioned above have been tested in clinical trials in different cancer types [7,8,9,10]. Despite preclinical optimism, the clinical trials resulted in only minor reductions in mortality, limiting clinical application of these P-gp inhibitors [11].

The anticancer agent doxorubicin is a highly potent compound used in cancer treatment, being one of the most widely used chemotherapeutics [12,13]. However, efficiency of dox treatment is limited due to low plasma solubility, poor blood pharmacokinetics, and non-selective cell killing, resulting in serious toxicity to healthy tissues, such as cardiac muscle [14]. In addition to these detrimental side effects, cancer cells exposed to doxorubicin often develop doxorubicin resistance [15]. Therefore, novel drug delivery systems to overcome drug resistance and off-target side effects are urgently needed. Motivated by these problems, we have designed a drug delivery system that can specifically deliver drug to the tumor site while simultaneously improving drug solubility and pharmacokinetics. As a result, a higher concentration of dox is delivered to the tumor, reducing the risk of harmful off-target effects, such as cardiotoxicity [16]. This drug delivery system consists of ELP–elastin-like polypeptide, CPP–cell penetrating peptide, a cleavable linker to enable doxorubicin release in the targeted low pH microenvironment of the tumor, and a derivative of the anticancer agent doxorubicin modified by a 6-maleimidocaproyl moiety for conjugation to a terminal cysteine residue on ELP. ELP is thermally responsive and improves the pharmacokinetics of this drug complex by reducing its clearance rate, while the CPP mediates cellular uptake of macromolecules. Doxorubicin is conjugated to ELP either through an acid sensitive linker or by the amino acid sequence Gly–Phe–Leu–Gly, (GFLG spacer) that serves as a substrate for lysosomal enzymes. Since the mechanism of ELP cellular uptake is endocytosis, ELP biopolymers would be expected to end up in lysosomes, where dox would be released due to the low pH and/or the action of lysosomal enzymes.

In this study, we compared cytotoxicity of cleavable (DOXO) and non-cleavable (ncDox) doxorubicin derivatives delivered by ELP biopolymers in drug-sensitive MCF-7 cancer cells and the drug-resistant cancer cell line NCI/ADR [17,18]. We tested the hypothesis that when ncDox is conjugated to the ELP biopolymer containing a lysosomally degradable GFLG spacer, the drug delivery construct will be more equally toxic to sensitive and resistant cell lines, compared to free dox. We expect the biopolymer constructs to deliver dox into cells by a mechanism that circumvents drug resistance. In order to characterize this drug delivery system, proliferation experiments were performed in dox-responsive and dox-resistant cell lines. We examined subcellular localization of dox using confocal microscopy. To investigate the mechanisms of action, we also measured cell cycle distribution and apoptosis using flow cytometry.

## 2. Results

### 2.1. Design of ELP-Based Drug Delivery Macromolecule

ELP-based biopolymer drug carriers were designed to deliver doxorubicin derivatives into cancer cells. A schematic of this drug delivery system is shown in Figure 1.

Due to its macromolecular size (60 kDa), ELP conjugation improves dox pharmacokinetics by reducing its clearance. Moreover, ELP is thermally responsive, which could allow this molecule to be targeted to the tumor site in vivo, by applying mild hyperthermia. Dox derivatives were attached to ELP by specific biochemical linkers, either an acid-cleavable hydrazone or an amino acid sequence (GFLG) that serves as a substrate for lysosomal enzymes. A cell penetrating peptide was included in the constructs to improve cellular uptake. SynB1 was chosen as the CPP because it has been shown to effectively deliver ELP conjugates to the cells with low inherent toxicity [19].

### 2.2. Comparison of Cytotoxic Effects of Several Drug Delivery Systems vs. Free Dox

Cytotoxicities of free dox and biopolymer drug delivery vehicles SynB1-ELP-DOXO and SynB1-ELP-GFLG-NCDox were measured by cell survival (Figure 2). After 72 h of treatment, IC50s (the concentration of drug required for 50% inhibition of proliferation) were determined (summarized in Table 1). Results indicate that the anti-proliferative effect of SynB1-ELP-DOXO is comparable to that of free doxorubicin in MCF-7 breast cancer cells. In doxorubicin-resistant NCI/ADR cells, SynB1-ELP-DOXO demonstrated high cytotoxicity, with an IC50 of 12.4 µM, nearly as toxic as free doxorubicin. While SynB1-ELP-GFLG-NCDox had higher IC50s compared to the other treatments in vitro, free dox required a 100-fold concentration increase to achieve the same toxicity in NCI/ADR compared to MCF7, but SynB1-ELP-GFLG-NCDox achieved equal toxicity in the resistant cell line with only a four-fold increase in concentration.

To compare the efficacy of the inhibition of proliferation between different constructs, proliferation experiments were repeated with dox equivalent concentrations of 0.6 and 1.7 µM for MCF-7, and 15 and 45 µM for NCI/ADR. Figure 3A shows that free doxorubicin and SynB1-ELP-DOXO have similar toxicity in MCF-7 cells, with SynB1-ELP-GFLG-NCDox showing less inhibition. In NCI/ADR cells (Figure 3B), treatment with free doxorubicin resulted in 34% survival with 15 µM drug and 9% at 45 µM, and the survival numbers for SynB1-ELP-DOXO were 47% and 34%, respectively. Survival percentages in MCF7 decreased proportionally as the doses increased in all three treatments. However, the difference in NCI/ADR cell survival between SynB1-ELP-GFLP-NCDox and SynB1-ELP-DOXO treatments narrowed substantially when the dose was increased from 15 to 45 µM, dropping from a 39% difference at 15 µM to a 22% survival difference at the higher dose.

### 2.3. Cellular Uptake of ELP-Delivered Doxorubicin

To quantify cellular uptake of ELP-delivered dox, we compared cellular association/uptake of free dox, SynB1-ELP-DOXO, and SynB1-ELP-GFLG-ncDox in MCF-7 and NCI/ADR cell lines (Figure 4). MCF7 and NCI/ADR cell lines were incubated with both ELP-dox delivery constructs at 2 μM dox equivalent concentration for 24 h. Doxorubicin is intrinsically fluorescent; thus, uptake levels were determined by dox fluorescence measured by flow cytometry.

Figure 4 shows histograms of total fluorescence for each of the treatments as well as auto-fluorescence from untreated controls. Figure 4B summarizes the amount of dox signal, and thus drug uptake, for each treatment after subtracting auto-fluorescence from total signal. Small, hydrophobic doxorubicin accumulated in cells to a greater extent than either of the biopolymer–drug conjugates, as would be expected.

### 2.4. Cell Cycle Distribution

One of the mechanisms of dox action is to inhibit topoisomerase II, resulting in DNA damage and cell cycle arrest [20]. In order to compare the effects of free dox, SynB1-ELP-DOXO, and SynB1-ELP-GFLG-ncDox on cell cycle distribution, MCF-7 and NCI/ADR cells were treated for 24 h with 2 µM dox equivalents and analyzed by flow cytometry (Figure 5).

As shown in Figure 5, after 24 h treatment with free Dox and SynB1-ELP-DOXO, most of the MCF-7 cells presented in G2 phase (free dox = 60%, SynB1-ELP-DOXO = 67%), followed by 35% in the G1 phase, and 5% of cells were in S phase. SynB1-ELP-GFLG-ncDox treatment did not seem to significantly alter the cell cycle pattern for either cell line. The percentage of cells in the G2/M phase cells was increased in MCF-7 treated with free dox and SynB1-ELP-DOXO. Only MCF-7 cells treated with free Dox and SynB1-ELP-DOXO showed accumulation of cells in G2/M. In contrast to MCF-7 cells, NCI/ADR cells showed a less distinct G0/G1 arrest after free dox treatment without a noticeable increase in G2 phase cells. However, treatment with SynB1-ELP-DOXO induced arrest in G2 phase after 24 h, suggesting a different mechanism of action. The SynB1-ELP-GFLG-ncDox construct alone does not significantly affect the cell cycle of NCI/ADR cells at the concentration and duration tested.

### 2.5. Intracellular Localization

The subcellular distribution of free dox, SynB1-ELP-DOXO, and Synb1-ELP-GFLG-ncDox was determined by confocal microscopy (Figure 6). Cells were incubated for 2 h with free dox, SynB1-ELP-DOXO, and SynB1-ELP-GFLG-ncDox at 25 μM Dox equivalent concentration.

Dox is a small, hydrophobic molecule that can easily pass the cell membrane, and it was found to be primarily distributed in nucleus in MCF-7 cells. NCI/ADR cells treated with free dox or SynB1-ELP-DOXO had both cytoplasmic and nuclear distribution of dox. SynB1-ELP-GFLG-ncDox showed cytoplasmic and peri-nuclear localization in both MCF-7 and NCI/ADR cell lines, suggesting a different mechanism of action than free dox.

### 2.6. Apoptosis

Apoptosis assay was performed with MCF-7 and NCI/ADR cell lines to examine mechanisms of toxicity. To measure apoptosis induced by biopolymer conjugated dox and free dox, cells were stained with Annexin V, which binds to phosphatidylserine on the outer leaflet of the plasma membrane and propidium iodide, which stains DNA. Flow cytometry experiments using a double staining Annexin/PI assay were performed to evaluate the induction of apoptosis by free dox, SynB1-ELP-DOXO, and SynB1-ELP-GFLG-ncDox in MCF-7 and NCI/ADR cell lines. Figure 7 shows that all treatments induce apoptosis. The apoptosis percentage of MCF-7 cells treated with SynB1-ELP-DOXO was 52.11%, while SynB1-ELP-GFLG-ncDOX induced 58.4% and free Dox 67.88%. In case of NCI/ADR, 19.47% of cells were apoptotic with SynB1-ELP-DOXO treatment, while free dox and SynB1-ELP-GFLG-ncDox-treated conditions exhibited 22.77% and 22.63% apoptotic cells, respectively. Besides the apoptotic cells, some necrotic cells were also observed, with the most necrotic cells present in the MCF-7 cell line treated with free dox. In contrast, treatment with biopolymer-bound dox reduced the percentage of cells positive for PI only (necrotic cells).

## 3. Discussion

Attachment of low-molecular weight anticancer therapeutics, such as doxorubicin, to macromolecular carriers results in drug delivery systems with numerous advantages, such as improved pharmacokinetic profiles, better solubility and opportunities for enhanced tumor targeting. In addition, coupling dox with macromolecular carriers can overcome P-gp-mediated dox resistance [21]. We reported earlier that thermal targeting of an acid-sensitive doxorubicin conjugate of elastin-like polypeptide enhances the therapeutic efficacy compared with the parent compound in vivo, confirming that dox delivered by the macromolecular carrier ELP has potential as a thermally targeted carrier for doxorubicin delivery [22,23]. In the current study, we used modified ELP constructs and demonstrated that dox delivered using an ELP-based polypeptide vector shows promise against dox-resistant cancer cells and warrants future study in vivo where the advantages of the drug delivery platform (enhanced solubility, diminished off-target effects, longer half-life in circulation, etc.) can be evaluated.

We compared cytotoxicity of cleavable (DOXO) and non-cleavable (ncDox) doxorubicin derivatives delivered by ELP in two cancer cell lines, MCF-7 and NCI/ADR. Here, we demonstrated dox delivered by ELP-based polypeptide carrier. SynB1-ELP-GFLG-ncDox can inhibit proliferation in dox-resistant cells, albeit less efficiently than free dox. This observation is consistent with our previous studies with similar ELP constructs in drug-sensitive human sarcoma cell line MES-SA, and its multidrug-resistant counterpart MES-SA/Dx5 [16]. The SynB1-ELP-GFLG-ncDox has a molecular weight of 60 kD, which requires that it enters cells via endocytosis, which is rate limited compared to free diffusion of small, hydrophobic dox molecules. The difference in toxicity in cell culture can be explained by the difference in the rate of cell uptake combined with the limitation of enzymatic cleavage rates within lysosomes to free the drug from the carrier. However, this difference in toxicity may diminish or even reverse in vivo, where drug pharmacokinetics and plasma solubility will play a role. In dox-resistant cells, we found a non-linear, disproportionate increase in inhibition with increased concentration of SynB1-ELP-GFLG-NCDox, suggesting that the mechanism of uptake and cellular retention of this construct may be less affected by drug resistance compared to free dox. Additionally, in dox-sensitive MCF-7 cells, the IC50 of the GFLG construct was two orders of magnitude greater than that of free dox, but in dox-resistant NCI/ADR cells, the IC50 value for SynB1-ELP-GFLG-NCDox fell within the same order of magnitude as the IC50 for free doxorubicin. These data suggest that SynB1-ELP-GFLG-NCDox gains some advantage in doxorubicin-resistant cells compared to its action in dox-sensitive cells.

Although SynB1-ELP-GFLG-ncDox is less potent than free dox in vitro, the macromolecular carrier offers many potential advantages that will be investigated in future animal studies. The most notable advantages include greater tumor accumulation and longer plasma half-life. ELP-based macromolecules have an additional advantage as drug carriers because they are thermally responsive, and this characteristic may be used to induce accumulation and targeting of the drug to locally heated sites [19]. Such enhancement in tumor targeting could effectively reduce the systemic dosage required to achieve therapeutic endpoints, thus sparing healthy tissues from some off-target effects.

Apoptosis is the preferred mechanism of cancer cell death in response to chemotherapy because necrosis induces an inflammatory response and is not desirable [24]. Therefore, the ability of these biopolymer drug conjugates to induce apoptosis in MCF-7 and NCI/ADR cell lines with limited necrosis is promising. The percentage of apoptotic cells treated with SynB1-ELP-GFLG-ncDox is higher than SynB1-ELP-DOXO treatment, which might be due to a different mechanism of action, particularly given the altered cellular localization profile. Since GFLG is a substrate for lysosomal enzymes, digestion of SynB1-ELP-GFLG-ncDox will result in dox, which would still be attached to two amino acids (LG-ncDox). It is not unreasonable that this molecule behaves differently, and those differences provide exciting avenues for future studies.

Multidrug resistance (MDR) is one of the most significant molecular mechanisms responsible for failure of chemotherapies [25]. Because MDR is mediated by the P-gp drug efflux pump, one of the approaches to reverse MDR has been to use P-gp inhibitors. First generation P-gp inhibitors have been used in clinics (cyclosporine A, quinidine, and verapamil). Second generation and third generation P-gp inhibitors have been developed that robustly inhibit P-gp. The best characterized third-generation P-gp inhibitors are elacridar, laniquidar, zosuquidar, ontogen, tariquidar [6]. Despite these tremendous advances, these drugs have yet to achieve significant therapeutic benefit for cancer patients. Mainly, the lack of clinical success stems from non-specific toxicity and often serious side effects. Additional approaches to overcoming MDR are urgently needed, and ELP-based biopolymer drug carriers, such as those investigated here, have previously been shown to circumvent P-gp pumps and overcome drug resistance [16]. In addition, other studies have demonstrated that ELP is a promising drug delivery tool for different chemotherapeutics with wide application in many cancers [23,26,27,28].

## 4. Materials and Methods

### 4.1. Polypeptide Expression and Purification

ELP based constructs, Synb1-ELP-GGC and SynB1-ELP-GFLG were designed and cloned via directional molecular cloning [29] and hyper-expressed [30] in E.coli BLR (DE3)-competent cells (Novagen, Madison, WI, USA). The polypeptides were then harvested and purified by the inverse thermal cycling as described previously [31].

### 4.2. Conjugation of Doxorubicin Derivatives to ELP

The ELP sequence was designed to contain a cysteine residue, which is then used for thiol-maleimide coupling of doxorubicin derivatives. We used two derivatives of doxorubicin: (i) DOXO, with acid-cleavable (6-maleimidocaproyl) hydrazone linker, and (ii) ncDox (derivative without acid-cleavable linker) that were designed and synthesized by Kratz et al. [27] (DOXO-EMCH, CytRx Pharmaceuticals, Freiburg, Germany). For maximizing the ELP–drug conjugation process and to avoid spontaneous formation of disulfide bonds causing undesirable protein self-aggregation, the following protocol was used. SynB1-ELP1-GGC or SynB1-ELP-GFLG protein at a concentration of 100 µM was solubilized in 50 mM sodium hydrogen phosphate (Na_2_HPO_4_) elution buffer, pH = 7.0, with addition of 10-fold molar excess (1 mM) of Tris (2-carboxyethyl) phosphine (TCEP) at room temperature for 30 min. Then, freshly prepared 800 µM doxorubicin derivatives were added to the solution and incubated for another 30 min at room temperature in the dark, followed by overnight incubation at 4 °C in the dark. Labeling efficiency and protein concentration were measured for absorbance at 280 and 495 nm, respectively. The protein–drug concentration was calculated as described [28].

### 4.3. Cell Lines

MCF-7 human breast cancer cell line and NCI/ADR human ovarian dox-resistant cancer cell line were purchased from ATCC (American Type Culture Collection). Cells were grown and maintained at 37 °C with 5% CO_2,_ and 95% humidity. Cells were cultured in Dulbecco’s modified Eagle’s minimum essential medium (DMEM) (Corning Inc., Corning, NY, USA), supplemented with 10% fetal bovine serum (FBS) (Atlanta Biologicals, Lawrenceville, GA, USA) and 1% penicillin–streptomycin antibiotics (HyClone Laboratories, Logan, UT, USA) To maintain log-phase growth, cells were passaged when they reached 70% confluency using 0.05% Trypsin (HyClone Laboritories, Logan, UT, USA) every two to three days.

### 4.4. Cytotoxicity Assay

To determine number of viable cells post treatment, the CellTiter-Glo^®^ assay was utilized according to the manufacturer’s instructions (Promega Corporation, Madison, WI, USA). Cells were seeded at 1 × 10^3^ cells in triplicate in opaque-walled 96-well plates. Cells were incubated overnight and treated with two-fold increasing concentrations of three different treatments as it follows: free doxorubicin, SynB1-ELP-DOXO and SynB1-ELP-GFLG-ncDox, and untreated control wells for 72 h. Plates were then screened for luminescence on a Synergy H4 plate reader (Agilent Technologies, Santa Clara, CA, USA) for quantification of live cells. Survival was quantified as signal percentage of untreated control cells.

### 4.5. Apoptosis Assay

Apoptosis induced by the drugs was determined by flow cytometry using a Gallios Flow Cytometer (Beckman Coulter, Brea, CA, USA). Briefly, MCF-7 and NCI/ADR cells were seeded at density of 1 × 10^5^ cells in 6-well plates. Cells were incubated overnight and then treated for 24 h with free dox, SynB1-ELP-DOXO, or SynB1-ELP-GFLG-ncDox at 2 µM dox equivalent concentration. For a positive control of apoptosis, cells were treated with etoposide at 500 µM concentration. At 24 h of treatment exposure, floating and attached cells were collected and assayed with PI/Annexin V double staining. Apoptotic positive cells were detected by fluorescence signal of Annexin V Alexa 488 binding to phosphatidylserine. Furthermore, we calculated the percentage of cells based on Annexin V Alexa 488 signal (early apoptosis)*,* Annexin V and PI signal (late apoptosis) and propidium iodide (PI)-only signal (necrosis) [32].

### 4.6. Cell Cycle Analysis

MCF-7 and NCI/ADR cells were plated in 6-well plates at density of 1 × 10^5^. After overnight incubation, cells were treated with free dox, SynB1-ELP-DOXO, and SynB1-ELP-GFLG-ncDox for 24 h at 37 °C. Then, treatment was removed, and the cells were washed with PBS and fixed with 3 mL ice cold 70% ethanol for 30 min. The cells were rinsed and resuspended in 500 μL PBS. To prevent any false positive signal from RNA, we added RNase A (Sigma-Aldrich, St. Louis, MO, USA) to a final concentration of 750 µg/mL to the cell suspension. For analyzing DNA content as an indicator of cell cycle progression, propidium iodide (PI) (Sigma-Aldrich, St. Louis, MO, USA) was added to the cell suspension at a concentration of 200 μg/mL and incubated for 30 min at room temperature. A Gallios flow cytometer (Beckman Coulter, Brea, CA, USA) was used to determine the intensity of PI fluorescence, and the results were analyzed with Kaluza software.

### 4.7. Cellular Localization

Cellular localization of ELP-delivered dox was determined by confocal microscopy. Briefly, MCF-7 and NCI/ADR cancer cells were plated at 50% confluence on 22 mm^2^ cover slips in a 6-well tissue culture plate. After 24 h incubation at 37 °C, media were removed, and cells were incubated for 2 h with 25 µM concentration of doxorubicin equivalent (free or ELP-bound). After 2 h of treatment, cells were washed with PBS, fixed with ice cold methanol, and stained with DAPI. Cells were then mounted on slides, sealed, and imaged with a Nikon confocal microscope (Nikon Instruments Inc., Melville, NY, USA).

### 4.8. Cellular Uptake Assay

MCF-7 and NCI/ADR cells were plated in 6-well tissue culture plates at density of 1.0 × 10^5^ and incubated at 37 °C overnight. Cells were treated for 24 h with free-Dox, SynB1-ELP-DOXO, or SynB1-ELP-GFLG-ncDox at a concentration of 2 µM dox equivalents. A non-enzymatic cell dissociation buffer was used to harvest the cells. The intrinsic dox fluorescence intensity (*n* = 10,000 cells) was measured using a Gallios flow cytometer and Kaluza software (Beckman Coulter, Brea, CA, USA). For analysis, cell debris and aggregates were gated with forward versus side scatter. All fluorescence intensity data were normalized to cellular auto-fluorescence.

### 4.9. Statistical Analysis

GraphPad Prism was used to graph and analyze cell survival data. Standard error of the mean (S.E.M.) was calculated from at least three independent experiments. A one-way ANOVA with Bonferroni tests for pair-wise comparison of treatment groups was performed to analyze the statistical differences between the treatment groups and the untreated control.

## 5. Conclusions

The results presented here demonstrate that ELP biopolymer drug carriers were able to induce apoptosis and inhibit proliferation of both dox-sensitive and resistant-cancer cells. Because ELP is thermally responsive and can be used as a thermally targeted drug carrier, these biopolymers have potential to specifically deliver doxorubicin to tumors in vivo and address MDR by circumventing efflux pumps. Furthermore, application of this approach can be extended to other small molecule drugs. The results obtained are encouraging and justify future studies to evaluate ELP biopolymers as a drug delivery platform in vivo.

## Figures and Tables

**Figure 1 ijms-23-02301-f001:**
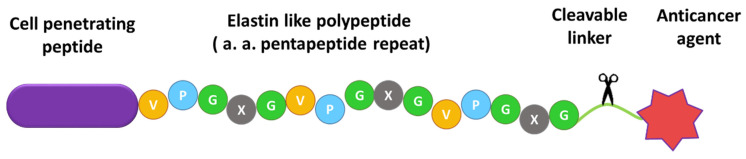
Conceptualization of the biopolymer drug delivery system. It consists of elastin-like polypeptide, cell penetrating peptide (SynB1), a cleavable linker to enable dox release in the targeted environment, and a derivative of doxorubicin.

**Figure 2 ijms-23-02301-f002:**
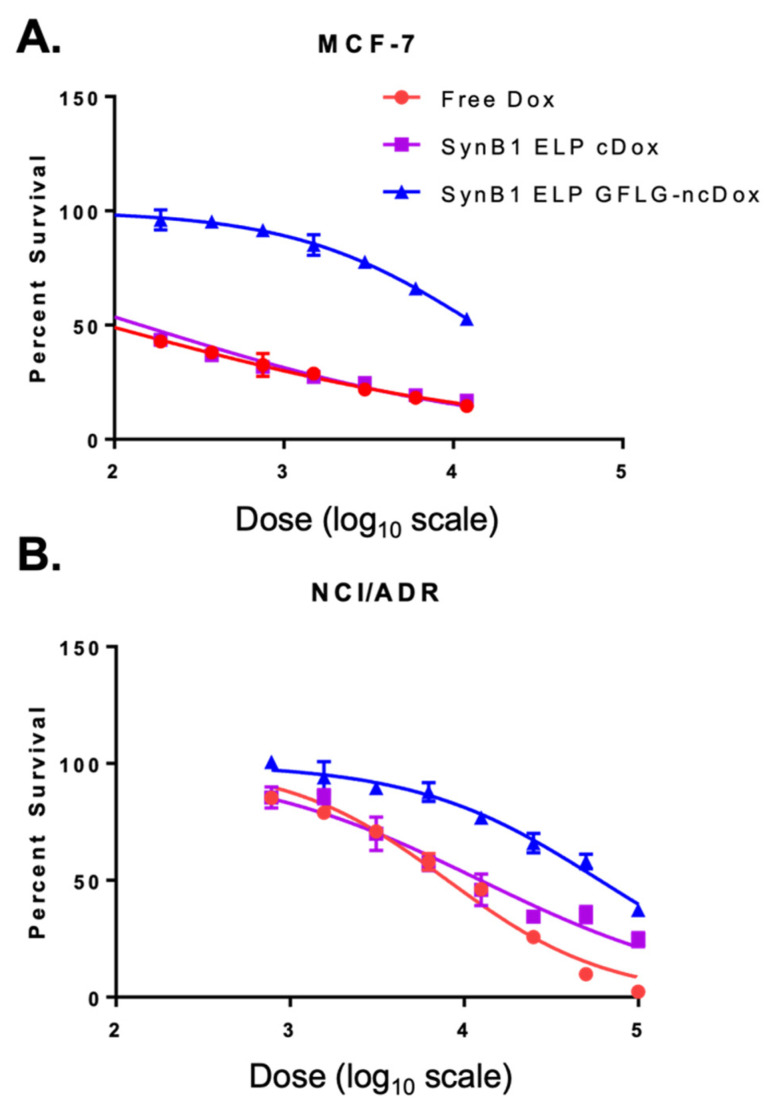
Cell viability assays in MCF-7 (**A**) and NCI/ADR (**B**) with variable doses of free doxorubicin, SynB1 ELP DOXO, and SynB1 ELP GFLG-ncDox. Doses (nM) have been converted to logarithmic scale (X-axis).

**Figure 3 ijms-23-02301-f003:**
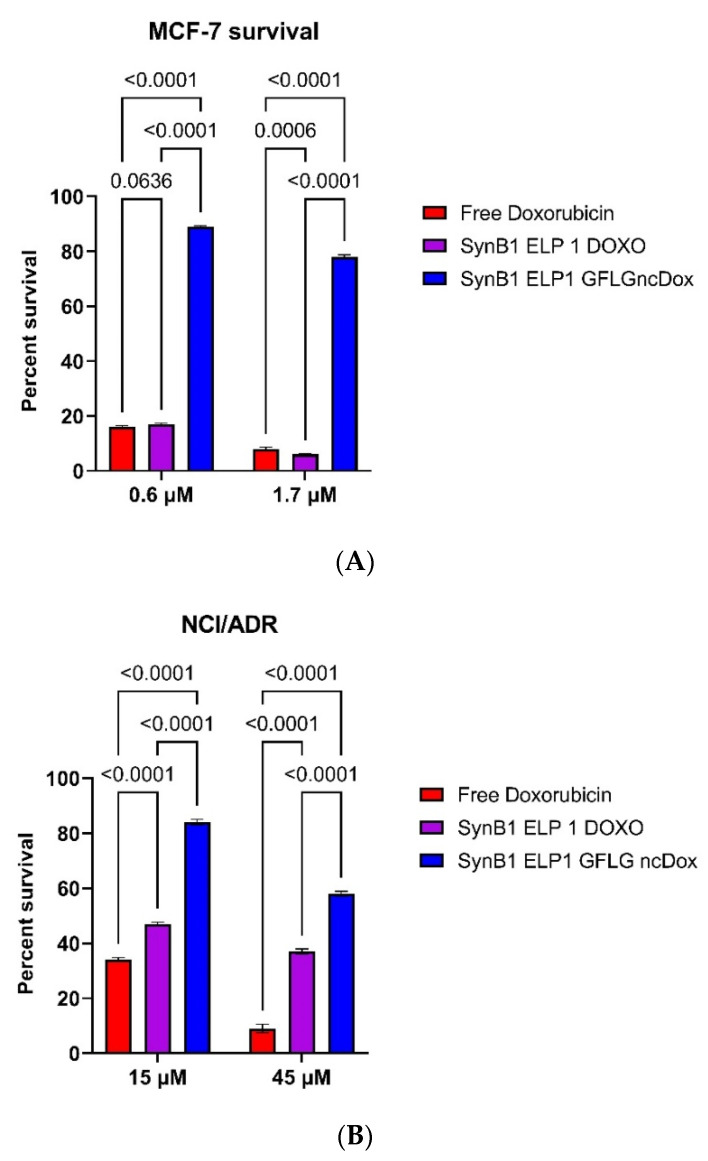
Cell viability assay. MCF-7 (**A**) and NCI/ADR (**B**) cells were treated with constructs carrying dox or free dox. Viability was determined on day 3 using the CellTitter Glo luminescent cell viability assay. One-way ANOVA with Bonferroni’s comparison test was performed using GraphPad Prism version, and *p*-values are shown.

**Figure 4 ijms-23-02301-f004:**
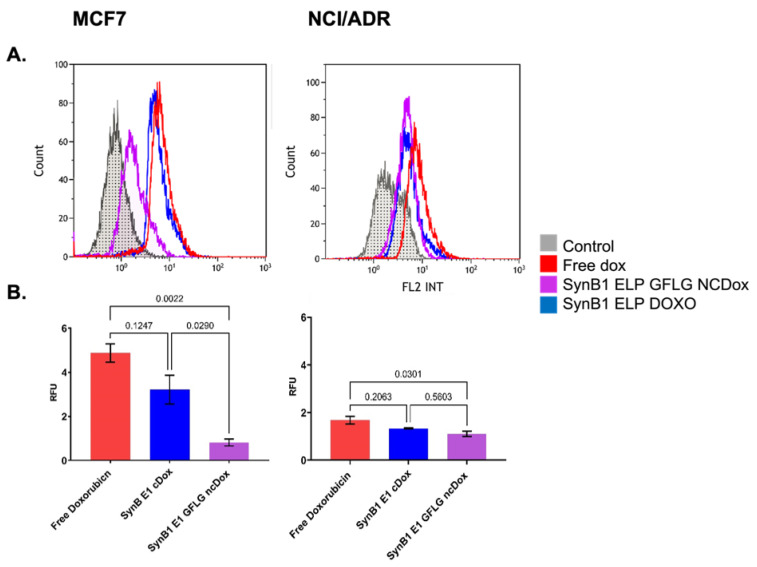
Cellular uptake of dox; all treatments at 2 µM dox equivalents for 24 h. (**A**) Representative histograms of doxorubicin uptake in MCF-7 and NCI/ADR cell lines are shown. *X*-axis shows fluorescence intensity at 575 ± 30 nm. Auto-fluorescence from untreated control cells depicted in gray. (**B**) Amount of total fluorescence contributed by dox. Data represent mean ± S.E.M. of at least three independent experiments.

**Figure 5 ijms-23-02301-f005:**
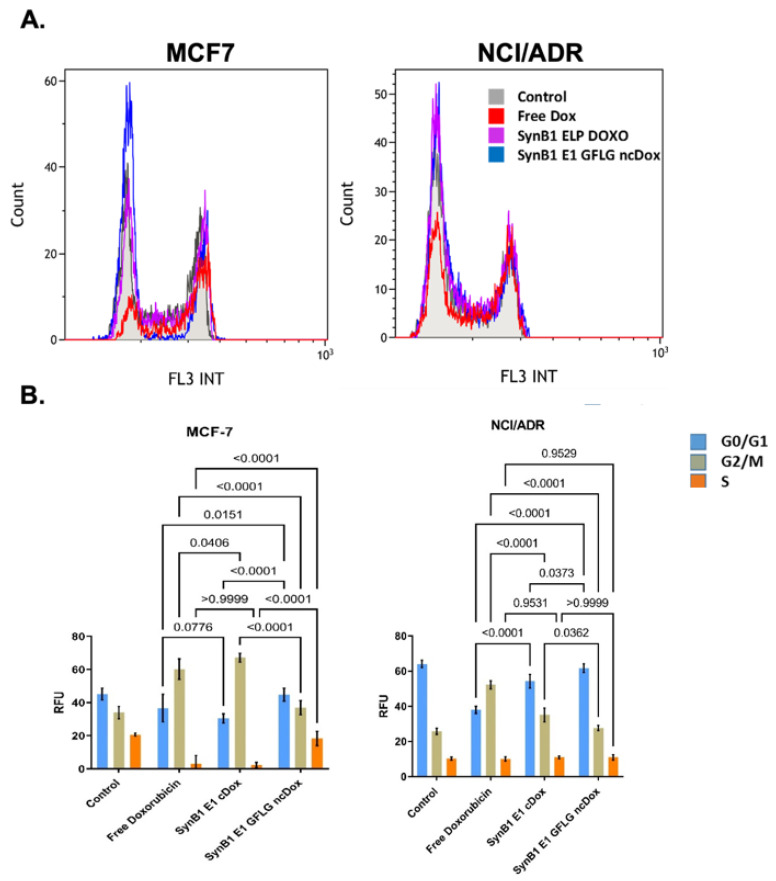
Cell cycle distribution. Cells were treated with 2 µM doxorubicin equivalents, incubated for 24 h and then analyzed using propidium iodide staining and flow cytometry. (**A**) Representative histograms of raw data. *X*-axis shows fluorescence intensity at 620 ± 30 nm. (**B**) Quantification of cells in each phase of the cell cycle. Data represent mean ± S.E.M of at least three independent experiments. *p*-values for differences between treatments and phases are shown.

**Figure 6 ijms-23-02301-f006:**
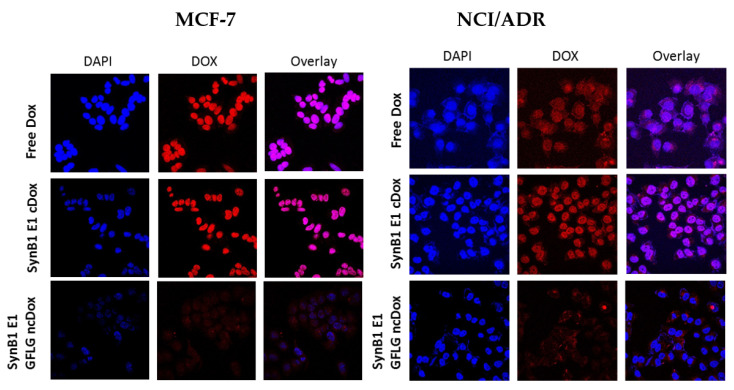
Confocal microscopy analysis of subcellular drug localization, all treatments at 25 µM dox equivalents for 2 h. Representative images of MCF-7 and NCI/ADR cells stained with DAPI (blue) to reveal nuclei. Doxorubicin fluorescence (red) shows intracellular localization of free dox, SynB1-ELP-DOXO, and SynB1-E1-GFLG-ncDox, and overlay (purple) represents the regions where doxorubicin has entered the nucleus. Images are qualitative only.

**Figure 7 ijms-23-02301-f007:**
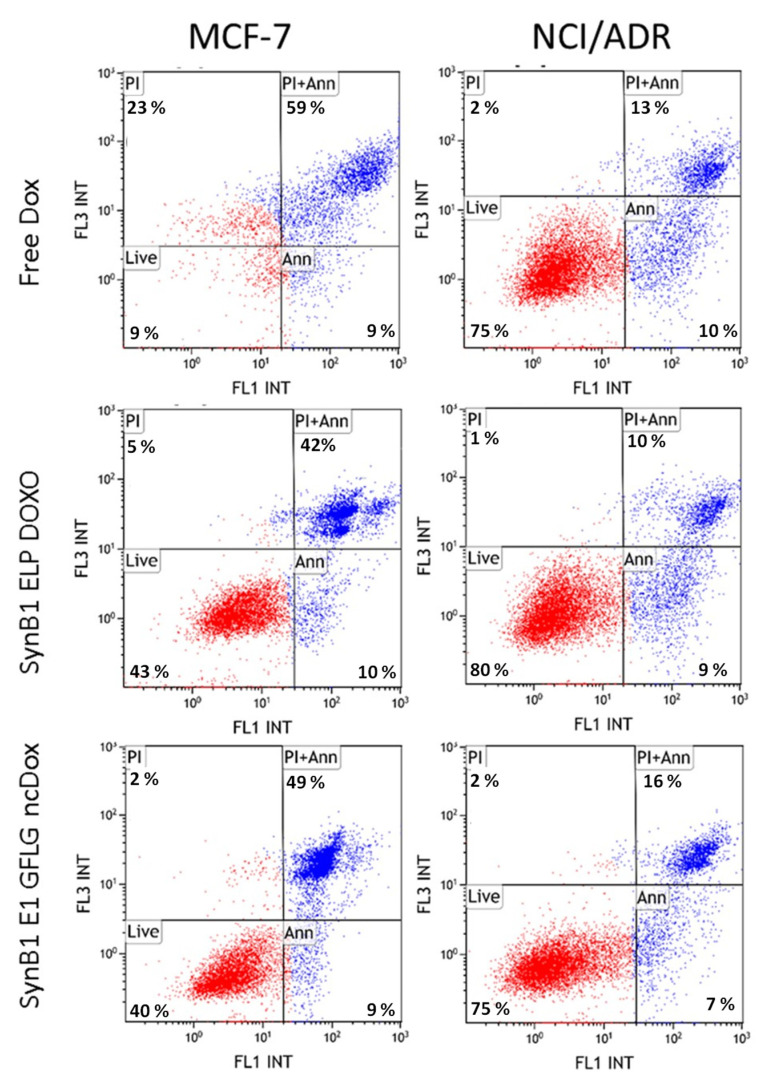
Induction of apoptosis by free dox, SynB1-ELP-DOXO, and SynB1-ELP-GFLG-ncDox in MCF-7 and NCI/ADR cell lines. Scatter plots show the live (lower left quadrant), early apoptosis (lower right quadrant), late apoptosis (upper right quadrant), and necrotic (upper left quadrant) MCF-7 and NCI/ADR cells after they were treated for 24 h with 2 µM dox-equivalent drug concentration. Percentage of apoptotic cells was determined based on gating for double staining with PI and Annexin V Alexa488.

**Table 1 ijms-23-02301-t001:** Comparison of IC50 Values (μM).

	MCF-7	NCI/ADR
Free Doxorubicin	0.089 ± 0.0182	8.029 ± 1.046
SynB1 ELP DOXO	0.143 ± 0.0387	12.414 ± 2.761
SynB1 ELP GFLG-NCDox	13.82 ± 1.638	60.169 ± 9.934

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
