# Peer review of "Circumventing Doxorubicin Resistance Using Elastin-like Polypeptide Biopolymer-Mediated Drug Delivery"

_ijms, 2022, doi:10.3390/ijms23042301_

Round 1

Reviewer 1 Report

The work is focused on an interesting topic - the search for new, more effective and safe, delivery forms of chemotherapeutic agents that may overcome drug resistance. However, the results presented are not convincing and manuscript is full of shortcomings. The manuscript in the present version is not suitable for publication.

Specific comments:

  1. The presented research shows that free doxorubicin is more toxic to the tested cancer cell lines than the biopolymer drug delivery vehicles SynB1-ELP- 203 DOXO and SynB1-ELP-GFLG-NCDox. What is the factual advantage of the tested constructs carrying Dox over a free drug? The suggested lower toxicity towards normal cells should be confirmed by an analogous experimental panel on normal cells.
  2. Statistical analysis has not been presented (no determination of statistic significance of the results, indication of statistical tests used, etc.). Adding this is necessary to draw conclusions.
  3. The Authors should standardize the provision on standard deviations – SEM or S.E.M. Why the Authors use in the manuscript SEM value despite the SD value?
  4. Some statements in the manuscript are not supported by the results, e.g. "Here we demonstrated that Dox delivered by ELP-based polypeptide carrier SynB1-ELP-GFLG-ncDox can efficiently inhibit proliferation in Dox resistant cells." The same about the title – it is not suitable for the presented results.
  5. The Authors stated: “To compare the efficacy of the inhibition of proliferation between different constructs, proliferation experiments were repeated with Dox concentrations 1.7uM and 0.6uM for MCF-7, and 45uM and 15uM for NCI/ADR).” The choice of the concentration should be precisely explained.
  6. Do results in figure 2 and figure 3 come from the same assay and experiment? If figure 3 is just another presentation of the results from the same experiment, then it should be included in one figure.
  7. Fig 4,5,7: in the figures or in their captions the concentration of the tested agents (dox equivalent concentration) should be given.
  8. Figure 7: the scatter plots of control should be presented.
  9. Figure 7: Cell density in the representative scatter plot of the sample: MCF-7 treated with free Dox appears to be significantly lower than in other samples. What is the reason?
  10. Figure 7 and Table 2 should have individual descriptions or percentages should be presented within the figure and not as a separate table.
  11. Figure 7 caption contains the description of the results – this must be moved to the main text.
  12. Figure 2 – designations A and B seem to be in a bad position.
  13. The manuscript is full of editorial errors.

Reviewer 2 Report

Dear authors, thank you, this is very interesting and important topic. The manuscript is readable, well arranged and offer compilation of a good results, congratulations.

Summing, I cannot find any drastic mistakes. This work is worth to be published in IJMS but:

  1. The novelty should be underlined clearly
  2. You should characterized your DDS fully, this could help answer the question “why does it works?” on molecular level.
  3. The conclusions need to be corrected strongly, as they are mostly “summary”

Author Response

Reviewer 2.

Dear authors, thank you, this is very interesting and important topic. The manuscript is readable, well arranged and offer compilation of a good results, congratulations.

Summing, I cannot find any drastic mistakes. This work is worth to be published in IJMS but:

  1. The novelty should be underlined clearly

Novelty is underlined clearly now

  1. You should characterized your DDS fully, this could help answer the question “why does it works?” on molecular level.

Discussion has been rewritten, and now includes the characterization of  molecular mechanism of drug delivery

  1. The conclusions need to be corrected strongly, as they are mostly “summary”

Discussion and conclusions have been extensively rewritten and corrected.

Two versions of the revised manuscript are included. One where all tracked changes are shown, and second version where all the changes are accepted

Round 2

Reviewer 1 Report

The Authors improved the manuscript and their answers are satisfactory for the most part. However, there is still few shortcomings:

  1. Statistical analysis has not been presented – in the revised version of the manuscript, there is still no determination of statistic significance of the results (basing on the p value) and the Authors did not give the information, which statistical tests were used. Adding this is necessary to draw conclusions.
  2. Figure 7: I suggest, that the scatter plots of control should be presented within the Figure 7 in the manuscript and the percentage data for individual quadrants should be given.

Reviewer 2 Report

Dear authors, I maintain my previous conclusion, this is a very interesting and important topic. The manuscript is readable, well arranged, and offers a compilation of good results, congratulations. Thank you for the corrections to my remarks.

Summing, I cannot find any drastic mistakes except technical ones:

  1. 2 and Table 1
  2. Physical remark: I’m not sure if you can use log(C) when C is concentration and has dimension – nM. Because then you should have axis dimension of something like log(nM); this is incorrect. It would be easier (and more readable) to use the logarithmic scale of concentration. Perhaps you don't feel the differences, but believe me, are huge.
  3. Philosophical question: Your results i.e. IC50 base on assumption that proliferation is in the range 100 – 0 %; what if this range is not correct?
  4. 4 please describe FL2 INT
  5. 5 please describe FL3 INT (where is FL1? ;)) please add 10^2 on the left side (or label any of the marks)
  6. Please add scale bars

Summing, this work, after corrections is worth to be published in IJMS.

Author Response

Dear authors, I maintain my previous conclusion, this is a very interesting and important topic. The manuscript is readable, well arranged, and offers a compilation of good results, congratulations. Thank you for the corrections to my remarks.

Summing, I cannot find any drastic mistakes except technical ones:

  1. 2 and Table 1
  2. Physical remark: I’m not sure if you can use log(C) when C is concentration and has dimension – nM. Because then you should have axis dimension of something like log(nM); this is incorrect. It would be easier (and more readable) to use the logarithmic scale of concentration. Perhaps you don't feel the differences, but believe me, are huge.

This is a good and helpful observation. These dose-response type curves are more ideally suited to in vivo situations where dose can be expressed as mg/kg or mg/m2. For in vitro experiments, where treatments are given in concentration using units with base-10 naming convention, we see how the axis dimension label was confounding. Converting the treatment concentrations before logarithmic conversion from nM to M would not change the shape of the curves but would simply shift them to the left of the y-axis. So the axis label has been changed in the manuscript, as has the caption for the figure to help clarify.

  1. Philosophical question: Your results i.e. IC50 base on assumption that proliferation is in the range 100 – 0 %; what if this range is not correct?

Interesting question. Because all treatment responses are only meaningful relative to controls (proliferation/signal in untreated wells acting as 100%), this 0-100% range is only applicable to data normalized to internal controls (which these are). In the future, it might be beneficial to add a positive control that is expected to kill every single cell, so there will be an internal control for minimal survival. However, background signal from DMSO-only wells was subtracted from all control and treatment measurements.

  1. 4 please describe FL2 INT & 5 please describe FL3 INT (where is FL1? ;)) please add 10^2 on the left side (or label any of the marks) 5 Please add scale bars

These X-axes show fluorescence intensity as measured through a given filter set (thus the numbers, which are specific to our cytometer (and probably other Gallios cytometers). We did not use a fluorophore which reports in the range of the FL1 filter. Additions to the figure captions have been made to clarify these labels.

Unfortunately, we cannot change the axis labels (tick marks) with the newer version of Kaluza software because it autoscales. We agree that it would be helpful to have these to better visualize scale, but, since these data are comparative, the units are relative anyway.

Round 3

Reviewer 1 Report

The manuscript still has deficiencies that can be corrected.

  1. In my opinion, it is necessary to present the determination of statistical significance for compared means in all bar graphs.
  2. The Authors replied, that they did not include representative scatter plots for control in Figure 7 because of the clarification. I disagree - adding controls and percent values for individual quadrants is needed by readers to read and interpret the results.

Author Response

We added statistical significance added to all bar chart and  corrected the flow cytometry chart as requested.

Thank you very much